# Simulation Study of FEUDT Structure Optimization and Sensitive Film Loading of SAW Devices

**DOI:** 10.3390/mi13101643

**Published:** 2022-09-30

**Authors:** Shen Bin, Haiyang Yang, Leiming Jiang, Xinlei Liu

**Affiliations:** School of Safety Engineering, Heilongjiang University of Science and Technology, Harbin 150022, China

**Keywords:** SAW gas sensor, forked-finger electrode, unidirectional forked-finger transducer, sensitive film, COMSOL

## Abstract

In order to further improve the degree of frequency response of the surface acoustic wave (SAW) sensor for gas detection, the structure of the forked-finger transducer was analyzed, and its optimal structural parameters were simulated and designed. The simulation model of the unidirectional fork-finger transducer is established by using COMSOL finite element software. The thickness of the piezoelectric substrate, the electrode structure and material, and the thickness of the coating film are optimized and simulated. The results show that: the optimal thickness of the piezoelectric substrate is 3*λ*. The optimal thickness ratio and the lay-up ratio of the forked-finger electrode are 0.02 and 0.5, respectively. The Al electrode is more suitable as the a forked-finger electrode material compared to Cu, Au and Pt materials. Under the same conditions, the metal oxide-sensitive film (ZnO and TiO_2_) has a higher frequency response than the polymer-sensitive film (polyisobutylene and polystyrene), and the best sensitive film thickness range is 0.5~1 μm.

## 1. Introduction

With the development of surface acoustic wave (SAW) technology, SAW devices are widely used in various fields for their small size, low insertion loss, and ease of wireless passivation. Currently, the finite element simulation technology of SAW devices is gradually maturing, and the best performance parameters can be obtained by adding multi-physics fields to implement simulation analysis. The forked-finger transducer structure is an important factor affecting the performance of SAW devices, and high-performance forked-finger transducers are a hot topic of discussion and research in recent years.

The unidirectional forked-finger transducer (UDT) structure of SAW devices, compared with the conventional bidirectional forked-finger transducer (IDT) structure, not only achieves the unidirectional propagation of acoustic waves, but also reduces the insertion loss of the device and improves the performance of SAW devices. Wang et al. constructed a delay-line transducer using a single-phase unidirectional transducer (SPUDT) that reduces device insertion loss [1]. Yang et al. conducted an analysis using finite element software and concluded that the structural parameters of IDTs have a significant impact on the performance of SAW devices [2]. J. X. Zhai and C. Chen et al. derived from finite element analysis that the FEUDT structure SAW sensor with six electrodes per cell effectively reduces the insertion loss of the SAW device compared to the IDT structure with two electrodes per cycle [3]. Jiaxin Zhai et al. studied the characteristics of the floating electrode unidirectional transducer (FEUDT) structure in detail using COMSOL finite element software, and the results showed that the FEUDT structure can excite SAW unidirectional propagation; the structure was applied to resonant and delayed line SAW gas sensors to achieve the highly sensitive detection of the gas concentration [4]. Kaixuan Li et al. also performed a comparative analysis of the characteristics of the IDT structure and FEUDT structure and concluded that the FEUDT structure not only has unidirectional propagation characteristics, but can also reduce the insertion loss of the device [5]. The above studies show that although the low-loss UDT structure of SAW sensors is more easily recognized and used by scholars compared to the IDT structure [6,7], for the UDT structure, it is possible to optimize the structural parameters to further reduce the generated losses; however this has not been explicitly outlined by scholars, which is the main issue studied in this paper.

The authors take the FEUDT structure, which is a special structure used in the UDT structure, as the object of study, and optimize the piezoelectric substrate, forked-finger electrode shape and electrode material of this structure using COMSOL finite element simulation software to derive the optimal design parameters. Additionally, based on the above optimal parameters, the law of coating film thickness is derived by analyzing the load-sensitive film of the device structure.

## 2. Modeling of SAW Sensors

### 2.1. Simulation and Experimental Discussion

There are two methods to analyze the performance of SAW sensors, one is the experimental method; the other is the simulation method. In this study, the best response frequency was obtained through the variation law of FEUDT structure parameters only via simulation experiments, and no fabrication or experimental verification was performed. However, in the actual application of IDT and UDT structures of the SAW transducer, it needs to be fabricated. In the fabrication process, it is not a two-dimensional model, but a geometric object. It is necessary to design not only the thickness, length and width of the substrate, but also the parameters such as the spacing of the pins, the height of the pins, the width of the finger strips of the electrodes, the spacing of the finger strips, the number of finger strips and the distance of the acoustic aperture. For resonator-type SAW sensors, a reflective grid must be designed and fabricated. In this process, not only is the manufacturing process strictly required, but the manufacturing time is also long. After the fabrication is completed, qualified SAW sensors are selected for testing experiments. The experimental analysis of the performance of SAW sensors with IDT and UDT structures shows the same results as the simulation analysis.

### 2.2. Mathematical Model Establishment

In this paper, we use the COMSOL finite element simulation and analysis software to construct a two-dimensional model of the SAW sensor and simulate its characteristics by adding solid mechanics and electrostatic physical fields. The solid mechanics equation is as follows [8].
(1)$$\frac{\partial \left(\frac{E}{1-\mu^{2}}\frac{\partial u}{\partial x}+\frac{\mu E}{1-\mu^{2}}\frac{\partial v}{\partial y}\right)}{\partial x}+\frac{\partial \left(G\left(\frac{\partial u}{\partial y}+\frac{\partial v}{\partial x}\right)\right)}{\partial y}=0$$
(2)$$\frac{\partial \left(\frac{E}{1-\mu^{2}}\frac{\partial v}{\partial y}+\frac{\mu E}{1-\mu^{2}}\frac{\partial u}{\partial x}\right)}{\partial y}+\frac{\partial \left(G\left(\frac{\partial u}{\partial y}+\frac{\partial v}{\partial x}\right)\right)}{\partial x}=0$$
(3)$$G=\frac{E}{2\left(1+\mu \right)}$$
where *E* is the elastic modulus, *μ* is the Poisson’s ratio, $\frac{\partial u}{\partial x}$ is the strain component in the *x* direction, $\frac{\partial v}{\partial y}$ is the strain component in the *y* direction, and *G* is the shear modulus.

Stress $\sigma_{ij}$ and electrical displacement $D_{i}$ are required to meet the following equations [9]:(4)$$\sigma_{ij}=c_{ijkl}^{E}\frac{\partial u_{k}}{\partial x_{k}},i,j,k,l=1,2,3$$
(5)$$D_{i}=e_{jkl}\frac{\partial u_{k}}{\partial x_{k}}-\varepsilon_{ik}^{s}\frac{\partial \varphi }{\partial x_{k}},i,j,k,l=1,2,3$$
where φ is the electric potential, $c_{ijkl}^{E}$ is the elastic coefficient, $e_{jkl}$ is the piezoelectric constant, and $\varepsilon_{ik}^{s}$ is the dielectric constant.

The periodic boundary conditions of the SAW device structure, constrained functions are:(6)$$U\left(x\right)=U\left(x+P\right)exp\left(-2\right)\pi \gamma n$$
(7)$$U\left(x\right)=\left[\begin{array}{c} u_{i}\left(x\right)\\ \varphi \left(x\right) \end{array}\right]$$
(8)γ=α+jβ
where $u_{i}\left(x\right)$ is the displacement of the surface acoustic wave, $\varphi \left(x\right)$ is the electric potential of the surface acoustic wave, γ is the complex propagation constant, α is the propagation loss, and β is the phase propagation constant.

### 2.3. Two-Dimensional Modeling

Compared to the conventional quartz piezoelectric substrate material, the author chose Y Z-LiNbO_3_ with a larger wave speed as the piezoelectric substrate material for SAW sensors [10,11], whose wave speed v is 3488 m/s and wavelength is denoted by *λ*. The expected center frequency of the SAW device is 146 MHz. Using its periodic structure, a periodic cell of the FEUDT structure was selected for two-dimensional modeling [12]. Its geometric parameters are shown in Table 1.

The electrodes are set for the FEUDT structure, which consists of six electrodes for one cycle cell. Electrodes 1 and 4 are connected to terminal and ground conditions, respectively; electrodes 2 and 5 impose a suspension potential condition; and electrodes 3 and 6 impose a zero surface charge density condition. This is shown in Figure 1.

The wavelength of SAW in the figure is represented by *λ*. The electrode center spacing *p* is equal to the sum of the electrode width *a* and the electrode gap *b*, and the size is *λ*/6.

The electrode materials used in SAW sensors are Al, Cu, Au and Pt. Au has better ductility and compatibility, and Al has lower acoustic impedance and better linear elasticity characteristics. In this paper, Al, Cu, Au and Pt are used as electrode materials for the analysis and comparison of resonant frequencies, and their parameters are shown in Table 2.

The boundary conditions of the SAW sensor model are set as follows: *Γ_T_* is a free boundary condition with zero point load applied; *Γ_B_* is a fixed constraint condition with zero point load applied; and *Γ_L_* and *Γ_R_* are a pair of periodic boundary conditions [4]. The schematic diagram of the two-dimensional model is shown in Figure 2a.

In the process of dividing the network of the model, the characteristics of SAW propagating mainly along the surface of the medium were considered. Therefore, the upper end of the model was densely divided, while the lower end was not densely divided to reduce the workload. The meshing diagram is shown in Figure 2b.

## 3. Results and Discussion

### 3.1. Modal Analysis

Modal is an inherent characteristic of SAW devices, and the author performed modal analysis by adding “eigenfrequency” studies based on the above parameters. The simulation results of SAW are divided into symmetric mode (a) and anti-symmetric mode (b) [13], as shown in Figure 3.

In order to determine the characteristic frequency of the SAW sensor, during the simulation, the physical field was controlled at a room temperature of 25 °C, and the AC excitation voltage was +12 V. The obtained resonance frequency (*f_sc+_*) and anti-resonance frequency (*f_sc−_*) are approximately 144 MHz and 147 MHz, respectively.
(9)$$f_{0}=\frac{\left(f_{sc+}+f_{sc-}\right)}{2}$$

The center frequency (*f*_0_) of the SAW sensor can be obtained using Formula (9), which is approximately 145.5 MHz, and the error is approximately 0.3%, which is basically consistent with the expected frequency.

From Figure 3, it can be seen that the energy of the SAW is mainly concentrated in the range of 1~2*λ* depths of the piezoelectric substrate, which meets the requirement of Rayleigh wave characteristics. Additionally, the darker the color in the surface area of the SAW sensor, the stronger the energy in that area, and the energy gradually decreases as the depth of SAW propagation within the piezoelectric substrate increases. Therefore, in order to ensure the accuracy of the analysis, 3~5*λ* of the piezoelectric substrate can be selected during the simulation and fabrication of the sensor.

In addition, by increasing the thickness of the piezoelectric substrate [14,15], the positive and negative resonant frequencies of the SAW device are also changed. With the electrode thickness kept at 0.5 μm and the electrode width kept constant at 2 μm, only the thickness of the piezoelectric substrate is changed to 1*λ*, 2*λ*, 3*λ*, 4*λ*, and 5*λ*, and the change of five sets of frequencies is measured, as shown in Figure 4.

Normally, the wave velocity of the acoustic surface wave is approximately 10% lower than that of the body wave, therefore, the acoustic surface wave can be expressed as [16]:(10)$$\xi =\beta \sqrt{1-{\left(\frac{v_{s}}{v_{b}}\right)}^{2}}\approx \frac{2\pi }{\lambda }\sqrt{1-{\left(\frac{9}{10}\right)}^{2}}\approx \frac{2.7}{\lambda }$$
where ξ is the attenuation coefficient of the sound surface wave, β is the wave number of the sound surface wave, $v_{s}$ is the wave speed of the sound surface wave, $v_{b}$ is the wave speed of the body wave, and *λ* is the thickness of the piezoelectric substrate.

It can be seen from Figure 4 that the positive and negative resonant frequencies generated by the four electrode materials show a decreasing trend with the increase in the piezoelectric substrate thickness, and the decreasing trend is basically the same. The main reason for this can be seen in Equation (10). As the thickness of the piezoelectric substrate increases, the degree of attenuation of the sound surface wave also becomes larger, and when its thickness is greater than 2*λ*, the attenuation coefficient is less than 1; that is, the energy of the SAW does not penetrate downward. Therefore, we chose the piezoelectric substrate thickness of 3*λ* for experimental analysis, which not only reduces the computational effort but also improves the simulation efficiency.

### 3.2. Thickness Ratio Analysis

Adding Al, Cu, Au, and Pt to the component materials, enabling one for each simulation and disabling the other three [17]. In the case of metallization rate *M_R_* = 0.5, the simulation range of thickness ratio *Q = h/**λ* is 0~0.09, and other parameters are shown in Table 1. Figure 5 shows the change in the resonance frequency and anti-resonance frequency with the thickness ratio under four electrode materials, namely Al, Cu, Au and Pt.

As can be seen in Figure 5, the positive and negative resonant frequencies of the SAW sensor decrease as the forked-finger electrode thickness increases, which leads to a lower detection accuracy of the sensor. On the contrary, when the electrode thickness is smaller, the performance of the SAW sensor is better, but this makes the electrode material prone to break. Without considering other factors, the optimal thickness ratio is selected here as 0.02, and the forked-finger electrode thickness is 0.5 μm.

### 3.3. Metallization Rate Analysis

On the basis of the thickness ratio analysis, the thickness of the electrode is selected as 0.5 μm. The metallization rate of the interdigital electrode is *M_R_ = a/p*, only the electrode width *a* is changed, and the electrode center distance *p* is kept unchanged. The range of the simulated metallization rate is 0~1. Figure 6 shows the change in the resonance frequency and anti-resonance frequency with the metallization rate under the four electrode materials of Al, Cu, Au and Pt.

It can be seen from Figure 6 that when *M_R_* = 0, the upper surface of the substrate has no forked-finger electrodes and is in a free boundary condition; when *M_R_* = 1, a layer of metal electrodes is deposited on the upper surface of the substrate and is in a metallization boundary condition. In both cases, there is no electrode mass effect in the SAW sensor; therefore, the positive and anti-resonant frequencies are equal. With the increase in the metallization rate, the more obvious the mass effect in the electrode, the greater the decrease of the positive and anti-resonant frequencies of the SAW sensor. On the contrary, the smaller the metallization rate, the smaller the electrode width, and the higher the detection accuracy of the SAW sensor, but this increases the difficulty of its manufacturing process. When the metallization rate is greater 0.5, the resonant frequency amplitude generated by the Al electrode changes less; therefore, 0.5 was chosen as the optimal metallization rate for this experiment.

### 3.4. Electrode Material Analysis

Al, Cu, Au and Pt were used as electrode materials to analyze and compare the resonance frequency [18]. The material parameters are shown in Table 2. The variation law of frequency is shown in Figure 7.

In Figure 7, under the same structural parameters, the higher the density of the metal electrode material, the higher the acoustic impedance, the greater the hindrance caused to the propagation of the SAW device, and the more serious the resulting energy loss. Compared with other metallic materials, Al materials have the advantages of a low cost, low density, and low acoustic impedance, and are more likely to be used in SAW sensors.

### 3.5. Film Thickness Analysis

According to the above parameters, the variation in different sensitive films with thickness is studied, and the influence of the frequency and wave speed of the SAW sensor is simulated and analyzed [19,20].

In this paper, two different of sensitive films, such as polymer polyisobutylene (PIB) film and polystyrene (PS) film, metal oxide zinc oxide (ZnO) film and titanium dioxide (TiO_2_) film, are taken as examples. The thickness of the simulated sensitive film ranges from approximately 0 to 3 μm. The results are shown in Figure 8 and Figure 9.

The relationship between effective wave speed (*v_s_*) and frequency is as follows:(11)$$v_{s\;}=\;f_{0}\;\times \;\lambda \;$$

As can be seen in Figure 8, the effective wave velocity of the acoustic surface wave changes according to the change in the frequency of the SAW sensor, taking the PIB-sensitive film as an example [21]. Due to the mass loading effect of the sensitive film, the greater the thickness of the sensitive film, the stronger the obstruction of the acoustic wave propagation by the SAW device, and the smaller its wave velocity. As can be seen in Figure 8, the response frequency increases as the thickness of the sensitive film decreases, and the propagation speed of SAW also increases. Therefore, the SAW sensor can effectively detect the physical properties of the gas using the coated film.

The relationship between the response (∆*f*) of the SAW sensor and the mass change (∆*m*) and the conductivity change (∆*σ_s_*) are shown in Equations (12) and (13), respectively [22].
(12)$$\Delta f=\Delta m\times \left(k_{1}+k_{2}\right)\times f_{0}^{2}$$
(13)$$\Delta f=-f_{0}\times \frac{K^{2}}{2}\times \left[\frac{1}{1+{\left[\frac{vc_{s}}{\Delta \sigma_{s}}\right]}^{2}}\right]$$
where *f*_0_ is the center frequency, ∆*m* is the mass change in the device’s sensitive film, *k*_1_ and *k*_2_ are the constants of the piezoelectric substrate, *v* is the wave speed, *K*^2^ is the electromechanical coupling coefficient, *c_s_* is the capacitance per unit length, and ∆*σ_s_* is the change in the conductivity of the film.

Additionally, as shown in Figure 9, for different sensitive films (PIB, PS and ZnO, TiO_2_), after 0.5 μm, the positive and negative resonant frequencies of the two types of sensitive films will gradually show different degrees of variation as the thickness of the sensitive film increases. Among them, the resonant frequency of metal oxide-sensitive film increases with the increase in film thickness, mainly because the mass and conductivity effects of metal oxide-sensitive film will affect the frequency change. The greater thickness of the metal oxide-sensitive film increases the electrical conductivity of the SAW sensor to a certain extent, which increases its frequency response. Therefore, for the above four sensitive materials, the thickness of the coated sensitive film should be greater than that of the forked-finger electrode, and the optimal sensitive film thickness range is within 0.5~1 μm to produce the best frequency response value.

## 4. Conclusions

In this work, a two-dimensional FEUDT structure model with a center frequency of 145 MHz was designed using COMSOL finite element simulation software with a low-power type FEUDT structure as the object of study for SAW gas sensors. Through the modal analysis, it was found that the energy of SAW is mainly concentrated in the range of 1~2*λ* of the piezoelectric substrate, and the optimal piezoelectric substrate thickness was selected to be 3*λ* to prevent the device performance form being affected. Increasing or reducing the thickness ratio of the forked-finger electrode and the metallization rate will have different degrees of influence on the device, and only when the thickness ratio and metallization rate are 0.02 and 0.5, respectively, is its influence at the minimal level. The metal Al with a higher frequency response, lower degree of influence, low cost and low acoustic impedance was selected as the forked-finger electrode material. The response performance of the SAW sensor is optimal only when the thickness of the coated sensitive film is higher than that of the forked-finger electrode, and the thickness of the sensitive film ranges from 0.5 to 1 μm. In our future work, we will focus more on the design and application of high frequency SAW gas sensors.

## Figures and Tables

**Figure 1 micromachines-13-01643-f001:**
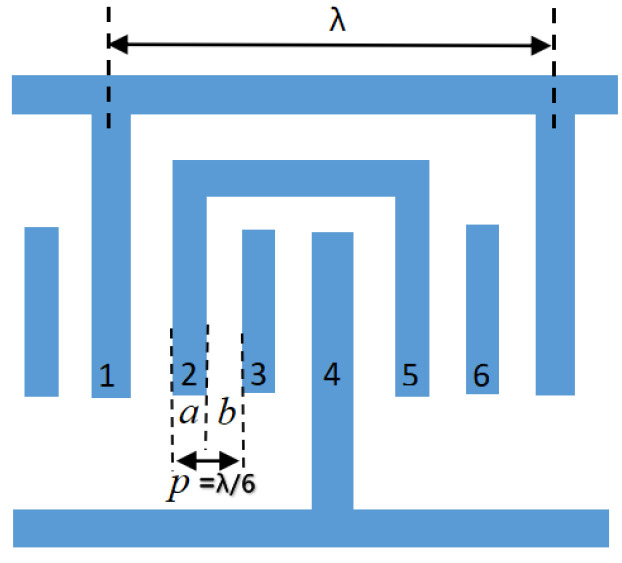
Schematic diagram of FEUDT structure.

**Figure 2 micromachines-13-01643-f002:**
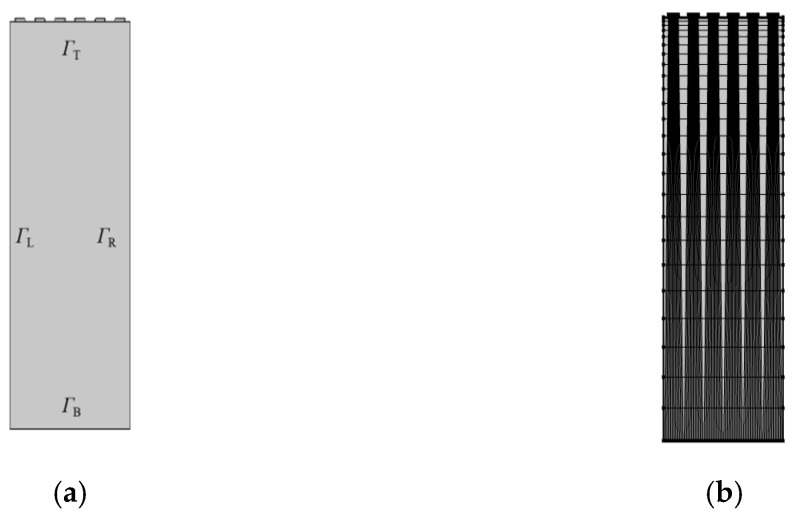
Schematic diagram of structure: (**a**) schematic diagram of the two-dimensional model; (**b**) schematic diagram of grid division.

**Figure 3 micromachines-13-01643-f003:**
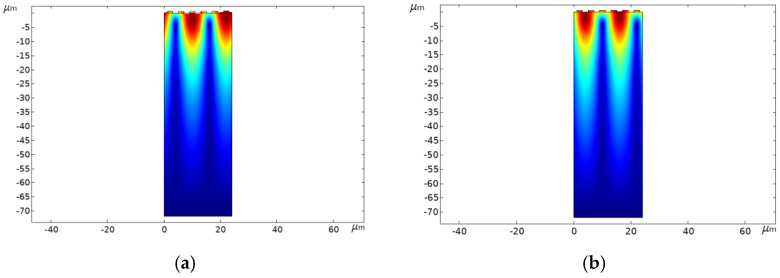
(**a**,**b**) Symmetrical and anti-symmetrical modes of Al electrode, respectively.

**Figure 4 micromachines-13-01643-f004:**
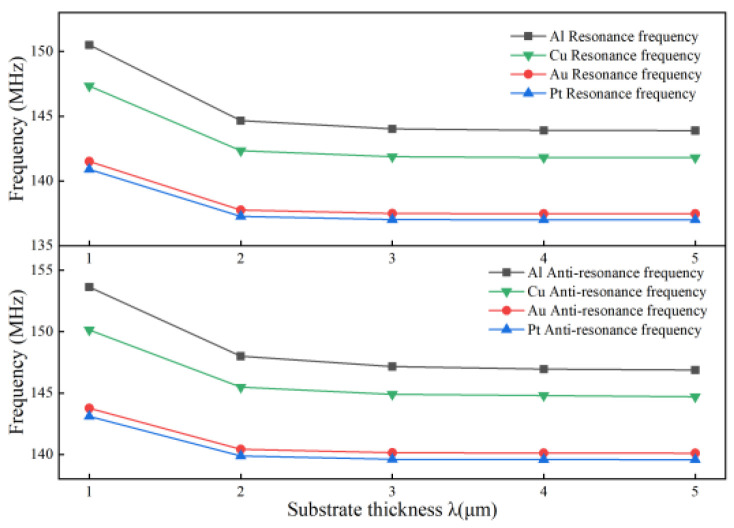
Substrate thickness versus frequency curve for different electrode materials.

**Figure 5 micromachines-13-01643-f005:**
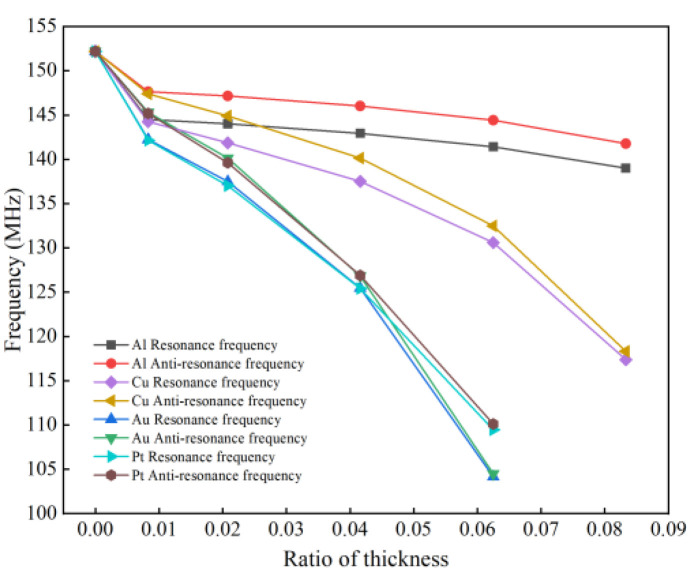
Variation characteristics of thickness ratio-frequency under different electrode materials.

**Figure 6 micromachines-13-01643-f006:**
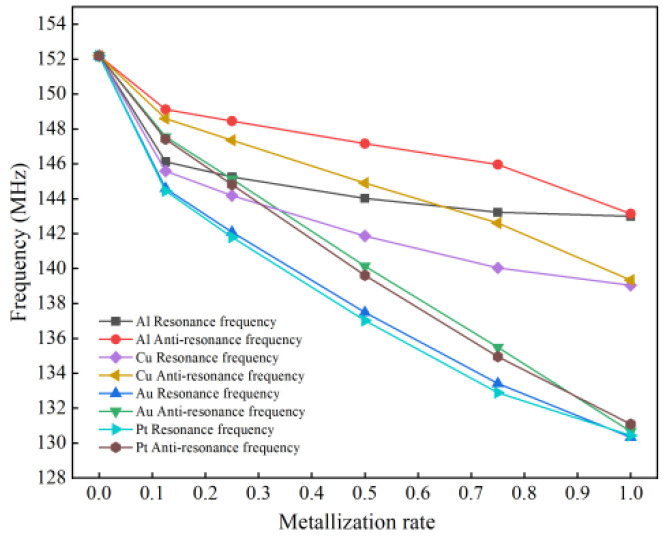
Variation characteristics of metallization rate-frequency under different electrode materials.

**Figure 7 micromachines-13-01643-f007:**
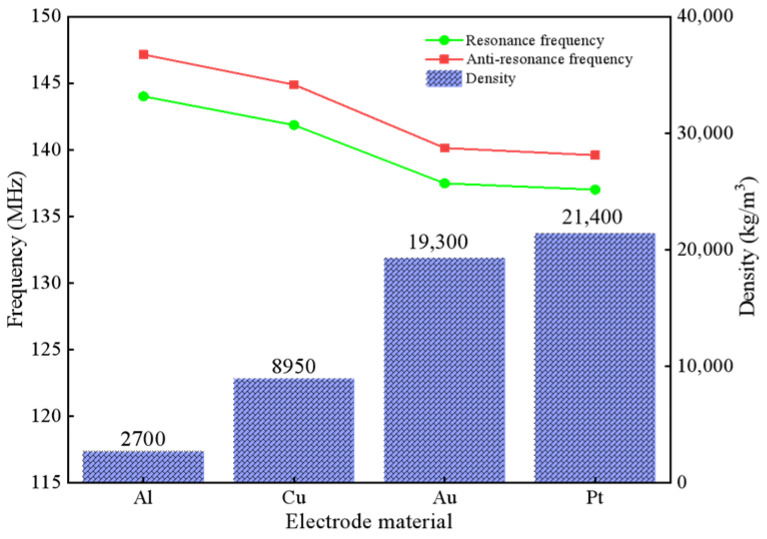
Different electrode materials-frequency characteristics.

**Figure 8 micromachines-13-01643-f008:**
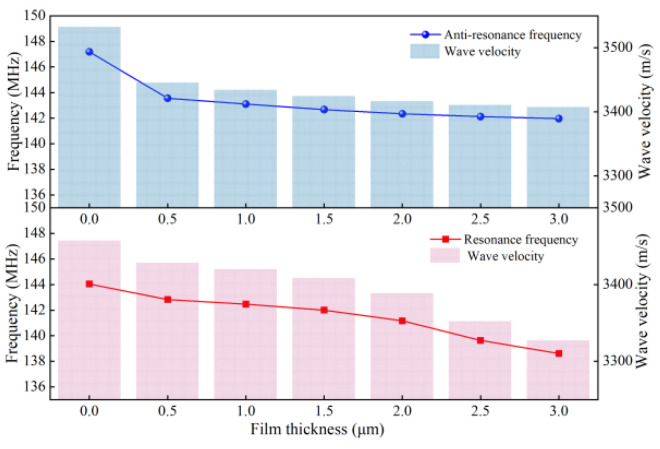
Thickness-frequency and wave velocity characteristics of PIB film.

**Figure 9 micromachines-13-01643-f009:**
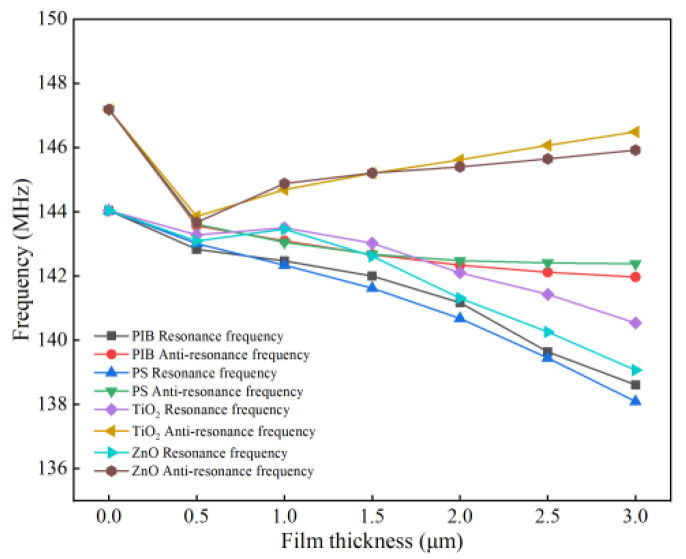
Thickness-frequency variation characteristics of different sensitive films.

**Table 1 micromachines-13-01643-t001:** Parameters of geometric model.

Variable	Parameter (μm)
Wavelength (*λ*)	24
Electrode center spacing (*p*)	4
Electrode width (*a*)	2
Electrode thickness (*h*)	0.5
Piezoelectric substrate thickness (*d*)	72

**Table 2 micromachines-13-01643-t002:** Material parameters of electrodes.

Variable	Al Parameter	Cu Parameter	Au Parameter	Pt Parameter
Young’s modulus *E*(10^10^ Pa)	7	11	7.95	16.9
Poisson ratio *μ*	0.33	0.326	0.42	0.38
Density *ρ* (10^4^ kg/m^3^)	0.27	0.895	1.93	2.14

## Data Availability

The data presented in this study are available on reasonable request from the corresponding author.

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
