# Peer review of "Simulation Study of FEUDT Structure Optimization and Sensitive Film Loading of SAW Devices"

_micromachines, 2022, doi:10.3390/mi13101643_

Round 1

Reviewer 1 Report

Tha last part of the last sentence in the abstract should be improved, please explain " the metal oxide sensitive film has a higher frequency response than the polymer sensitive film, and the best sensitive film thickness range is 0.5~1 μm." - the frequency response depends on the used analyte compounds - so please change this too general conclusion.

The FEUDT abbreviation also should be somwhere explained.

In the Table 2 (improve the number) the number powers (i.e. 1010) should be in only in the first column not in every ones. Besides the elasticity modulus is written as Young not Yang.

Please explain the sentence in 200 and 201 :  "It is important to note that if the coated sensitive film is too thin, the adsorption effect on the gas is not significant." - general the SAW device is especially good just for a thin films when for instance the resistance method cannot be applied - pls explain 

Also it is not clear how the simulations were performed just for the sensitive various films, i.e. what parameters of these materials are utilized for simulations.

Author Response

Response to Reviewer 1 Comments

Point 1: The last part of the last sentence in the abstract should be improved, please explain " the metal oxide sensitive film has a higher frequency response than the polymer sensitive film, and the best sensitive film thickness range is 0.5~1 μm." - the frequency response depends on the used analyte compounds - so please change this too general conclusion.

Response 1: The "the metal oxide sensitive film has a higher frequency response than the polymer sensitive film, and the best sensitive film thickness range is 0.5~1 μm." is not strict, and has been revised to " Under the same conditions, the metal oxide-sensitive film (ZnO and TiO2) has a higher frequency response than the polymer-sensitive film (polyisobutylene and polystyrene), and the best sensitive film thickness range is 0.5~1 μm." and marked in the abstract.

Point 2: The FEUDT abbreviation also should be somwhere explained.

Response 2: The abbreviation for floating electrode unidirectional transducer is FEUDT,which has been modified and labeled in the introduction out.

Point 3: In the Table 2 (improve the number) the number powers (i.e. 1010) should be in only in the first column not in every ones. Besides the elasticity modulus is written as Young not Yang.

Response 3: The numerical powers in Table 2 have been placed in the first column and the elasticity modulus  has been written as Young, both to be labeled in Table 2.

Point 4: Please explain the sentence in 200 and 201 : "It is important to note that if the coated sensitive film is too thin, the adsorption effect on the gas is not significant." - general the SAW device is especially good just for a thin films when for instance the resistance method cannot be applied - pls explain 

Response 4: The phrase "It is worth noting that" is not well formulated and should be revised to "As can be seen in Figure (8), the response frequency increases as the thickness of the sensitive film decreases, and the propagation speed of SAW also increases. Therefore, the SAW sensor can effectively detect the physical properties of the gas using the coated film". Marked in the text.

Point 5: Also it is not clear how the simulations were performed just for the sensitive various films, i.e. what parameters of these materials are utilized for simulations.

Response 5: The author only simulated and analyzed two materials: metal oxide sensitive film and polymer sensitive film. In COMSOL finite element software, there are various common sensitive materials in the "material" library, so select the ones you need to add. In the material properties, the parameters such as density, Young’s modulus, Poisson ratio and sound velocity are mainly set, and these parameters are determined by searching the data, while the rest of the parameters are the default values.

Reviewer 2 Report

The manuscript discusses a simulation model of the unidirectional fork-finger transducer using COMSOL finite element software. The thickness of the piezoelectric substrate, the electrode structure and material, and the thickness of the coating film are optimized and simulated by adding the coupled physical fields of solid mechanics and electrostatics. The authors revealed that the optimal thickness of the piezoelectric substrate is 3λ; the optimal thickness ratio and the lay-up ratio of the fork-finger electrode are 0.02 and 0.5, respectively. The Al electrode is more suitable as the fork-finger electrode material compared with Cu, Au, and Pt materials; the metal oxide sensitive film has a higher frequency response than the polymer sensitive film, and the best sensitive film thickness range is 0.5~1 μm.

Comments:

1.       Overall, the knowledge and the simulation analysis presented represent suitable and valuable information and work regarding the FEUDT structure optimization and exploring the sensing film load. the figures and references listed have been cited in the contents, and the discussion of related work and associated references are adequate.

2.       The paper is well structured and written. The Abstract and the introduction sections provide helpful information for the readers. Nevertheless, I believe that the abstract is relatively long. I recommend shortening and focusing on the Abstract. For example, in the abstract, the sentence “The SAW gas sensor is a microelectromechanical system device that can detect trace amounts of toxic and hazardous gases and make alarms” can be omitted.

3.       The Introduction as reported is a full summary of SAW devices and the IDT and UDT which is well but does not highlight the aim work of the author. I recommend that the end of the introduction should be revised.

4.       The model was studied by a finite element method (FEM) modeling using the Comsol Multiphysics”.

a.       There are many versions of COMSOLMULTIPHYSICS which was used for such studies.

b.       The model is strongly simplified. The authors present two-dimensional modeling. It is necessary to reveal all important simplifications. For instance, it ignores the fact of piezoelectric surface metalization (depending on the number of electrodes and shape of ITD). The metallization can change the SAW velocity in a significant way. I recommend that the authors justify their choice of modulation.

5.       The FEM approach to SAW research is quite interesting and may deliver much important information about the potential properties of the devices without the necessity of manufacturing and experimental verification. However, it is recommended that the authors add a short paragraph highlighting some points and discussion about the experimental work and comparison, when applicable, for the IDT and UDT.

6.       The English are not completely satisfactory. It needs improvement. Below are a few examples of typos and nitpicking linguistic remarks:

                                 I.            Use short sentences with “full stops” for example in the Abstract

The simulation model of the unidirectional fork-finger transducer is established by using COMSOL finite element software. The thickness of the piezoelectric substrate, the electrode structure and material, and the thickness of the coating film are optimized and simulated by adding the coupled physical fields of solid mechanics and electrostatics. The results show that: the optimal thickness of the piezoelectric substrate is 3λ.  The optimal thickness ratio and the lay-up ratio of the fork-finger electrode are 0.02 and 0.5, respectively; 15 the Al electrode is more suitable as the fork-finger electrode material compared with Cu, Au, and Pt 16 materials. The metal oxide sensitive film has a higher frequency response than the polymer sensitive film, and the best sensitive film thickness range is 0.5~1 μm.

                               II.            Figure 3.Figure 3. Symmetrical and anti-symmetrical modes of Al electrode: (a) Symmetrical; (b) 109 Anti-symmetrical. Can be replaced as Figure 3. (a) and (b) Symmetrical and anti-symmetrical modes of Al electrode, respectively.

                             III.            Page 5: Adding Al, Cu, Au, and Pt to the components' materials, enabling…..

                             IV.            Page 6: line 170 After the metallization rate of 0.5 can be replaced by beyond …………

7.       This is a good paper, but I have mixed feelings regarding it. There is very successful simulation work, but on the other side, compared to some experimental work and no new physics, electronics, or science explanations were learned, and the primary motivation or the main objective is not clear. The authors can add some clear sentences and discussion to address the main motivation and the message of the results and their comparison to the experimental work.

a.       For example, on page 5, the authors reported from Figure 4 that “Before the substrate thickness of 2λ, the frequency variation is large, and in the range of 3~5λ, the frequency remains basically the same” what is the author’s interpretation of this finding?

b.       Additionally, on page 6, the authors reported from Figure 6 that” when MR=0 and MR=1, there is no electrode mass effect in the SAW sensor, so that the positive and antiresonant frequencies are equal”. kindly give discussion and justifications

Author Response

Response to Reviewer 2 Comments

Point 1: Overall, the knowledge and the simulation analysis presented represent suitable and valuable information and work regarding the FEUDT structure optimization and exploring the sensing film load. the figures and references listed have been cited in the contents, and the discussion of related work and associated references are adequate.

Point 2: The paper is well structured and written. The Abstract and the introduction sections provide helpful information for the readers. Nevertheless, I believe that the abstract is relatively long. I recommend shortening and focusing on the Abstract. For example, in the abstract, the sentence “The SAW gas sensor is a microelectromechanical system device that can detect trace amounts of toxic and hazardous gases and make alarms” can be omitted.

Response 2: The first sentence of the abstract "The SAW gas sensor is a microelectromechanical system device that can detect trace amounts of toxic and hazardous gases and make alarms" has been deleted, and "by adding solid mechanics and electrostatic coupling of physical fields" has been deleted. The summary was shortened appropriately, and marked it in the abstract.

Point 3: The Introduction as reported is a full summary of SAW devices and the IDT and UDT which is well but does not highlight the aim work of the author. I recommend that the end of the introduction should be revised.

Response 3: The issue of not highlight the aim work of the author in the end of introduction has been modified to read ""The above studies show that although the low-loss UDT structure of SAW sensors is more easily recognized and used by scholars compared to the IDT structure [6,7], for the UDT structure, it is possible to optimize the structural parameters to further reduce the generated losses; however is has not been explicitly outlined by scholars, which is the main issue studied in this paper. The authors take the FEUDT structure, which is a special structure used in the UDT structure, as the object of study, and optimize the piezoelectric substrate, forked-finger electrode shape and electrode material of this structure using COMSOL finite element simulation software to derive the optimal design parameters. Additionally based on the above optimal parameters, the law of coating film thickness is derived by analyzing the load-sensitive film of the device structure." , and is marked at the end of the introduction.

Point 4: The model was studied by a finite element method (FEM) modeling using the Comsol Multiphysics”.

          a.There are many versions of COMSOLMULTIPHYSICS which was used for such studies.

Response 4 (a): This paper used Comsol Multiphysics 5.6.

           b.The model is strongly simplified. The authors present two-dimensional modeling. It is necessary to reveal all important simplifications. For instance, it ignores the fact of piezoelectric surface metalization (depending on the number of electrodes and shape of ITD). The metallization can change the SAW velocity in a significant way. I recommend that the authors justify their choice of modulation.

Response 4 (b): Based on the periodic structure of the forked-finger transducer in the SAW device, periodic boundary conditions are used for the two-dimensional model, for the boundary of the upper surface of the piezoelectric substrate, the mechanical boundary condition is imposed as a free boundary condition, and the electrical boundary condition is imposed as a zero-charge boundary condition, for the boundary of the lower surface of the piezoelectric substrate, the mechanical boundary condition is imposed as a fixed constraint boundary condition, and the electrical boundary condition is imposed as a zero-charge boundary condition, for the left and right boundaries of the piezoelectric substrate, both mechanical and electrical boundary conditions was imposed as periodic boundary conditions. And the number of fingers of the electrodes is so large that it is not necessary to use all of them in the simulation. The forked-finger transducer of one periodic unit is chosen instead of the whole device for simulation, which not only reduces the computational effort but also improves the efficiency of the simulation. Moreover, this simulation is to design the parameters of the forked-finger under the most ideal conditions, rather than manufacturing and application, so the actual application effect needs to be verified.

Point 5: The FEM approach to SAW research is quite interesting and may deliver much important information about the potential properties of the devices without the necessity of manufacturing and experimental verification. However, it is recommended that the authors add a short paragraph highlighting some points and discussion about the experimental work and comparison, when applicable, for the IDT and UDT.

Response 5: A short paragraph has been added, for subsection 2.1,“There are two methods to analyze the performance of SAW sensors, one is the ex-perimental method; the other is the simulation method. In this study, the best response frequency was obtained through the variation law of FEUDT structure parameters only via simulation experiments, and no fabrication or experimental verification was performed. However, in the actual application of IDT and UDT structures of the SAW transducer, it needs to be fabricated. In the fabrication process, it is not a two-dimensional model, but a geometric object. It is necessary to design not only the thickness, length and width of the substrate, but also the parameters such as the spacing of the pins, the height of the pins, the width of the finger strips of the electrodes, the spacing of the finger strips, the number of finger strips and the distance of the acoustic aperture. For resonator-type SAW sensors, a reflective grid must be designed and fabricated. In this process, not only is the manufacturing process is strictly required, but the manufacturing time is also long. After the fabrication is completed, qualified SAW sensors are selected for testing experiments. The experimental analysis of the performance of SAW sensors with IDT and UDT structures shows the same results as the simulation analysis.” This paragraph has been added to page 2 and is marked.

Point 6: The English are not completely satisfactory. It needs improvement. Below are a few examples of typos and nitpicking linguistic remarks:

           I. Use short sentences with “full stops” for example in the Abstract

The simulation model of the unidirectional fork-finger transducer is established by using COMSOL finite element software. The thickness of the piezoelectric substrate, the electrode structure and material, and the thickness of the coating film are optimized and simulated by adding the coupled physical fields of solid mechanics and electrostatics. The results show that: the optimal thickness of the piezoelectric substrate is 3λ.  The optimal thickness ratio and the lay-up ratio of the fork-finger electrode are 0.02 and 0.5, respectively; 15 the Al electrode is more suitable as the fork-finger electrode material compared with Cu, Au, and Pt 16 materials. The metal oxide sensitive film has a higher frequency response than the polymer sensitive film, and the best sensitive film thickness range is 0.5~1 μm.

Response 6 (I): The use of phrases with "full stops" in the abstract has been adjusted and has been marked in the abstract.

           II.Figure 3.Figure 3. Symmetrical and anti-symmetrical modes of Al electrode: (a) Symmetrical; (b) 109 Anti-symmetrical. Can be replaced as Figure 3. (a) and (b) Symmetrical and anti-symmetrical modes of Al electrode, respectively.

Response 6 (II): "Figure 3. Symmetrical and anti-symmetrical modes of Al electrode: (a) Symmetrical; (b) Anti-symmetrical". has been replaced as "Figure 3. (a) and (b) Symmetrical and anti-symmetrical modes of Al electrode, respectively." and has been marked.

           III. Page 5: Adding Al, Cu, Au, and Pt to the components' materials, enabling…..

Response 6 (III):  On page 5, the sentence "Add Al, Cu, Au, and Pt to the components' materials, enabling one for each simulation and disabling the other three" has been changed to "Adding Al, Cu, Au, and Pt to the components' materials, enabling one for each simulation and disabling the other three." by touching up, and has been marked.

              IV. Page 6: line 170 After the metallization rate of 0.5 can be replaced by beyond …………

Response 6 (IV.): "After metallization rate 0.5" has been replaced with "When metallization rate is greater 0.5", and has been marked.

Point 7: This is a good paper, but I have mixed feelings regarding it. There is very successful simulation work, but on the other side, compared to some experimental work and no new physics, electronics, or science explanations were learned, and the primary motivation or the main objective is not clear. The authors can add some clear sentences and discussion to address the main motivation and the message of the results and their comparison to the experimental work.

              a.For example, on page 5, the authors reported from Figure 4 that “Before the substrate thickness of 2λ, the frequency variation is large, and in the range of 3~5λ, the frequency remains basically the same” what is the author’s interpretation of this finding?

Response 7 (a): Added a new equation on page 5, explained the meaning of the variables in it, and revised "The frequency varies greatly until the substrate thickness of 2λ, and remains essentially constant in the range of 3~5λ" to "The main reason for this can be seen in Equation (10). As the thickness of the piezoelectric substrate increases, the degree of attenuation of the sound surface wave also becomes larger, and when its thickness is greater than 2λ, the attenuation coefficient is less than 1, that is, the energy of the SAW does not penetrate downward.", and marked it .

            b.Additionally, on page 6, the authors reported from Figure 6 that” when MR=0 and MR=1, there is no electrode mass effect in the SAW sensor, so that the positive and antiresonant frequencies are equal”. kindly give discussion and justifications

Response 7 (b): The expression of this sentence is not strict enough, and the phrase " when MR=0 and MR=1, there is no electrode mass effect in the SAW sensor, so that the positive and antiresonant frequencies are equal" has been revised to "When MR=0, the upper surface of the substrate has no forked-finger electrodes and is in a free boundary condition; when MR=1, a layer of metal electrodes is deposited on the upper surface of the substrate and is in a metallization boundary condition. In both cases, there is no electrode mass effect in the SAW sensor; therefore, the positive and anti-resonant frequencies are equal", and marked it on page 6.

Reviewer 3 Report

The design and method and conclusion of this paper have been reported previously. the author shold find some point what is the problem they want to solve

Author Response

Response to Reviewer 3 Comments

Point 1: The design and method and conclusion of this paper have been reported previously. the author shold find some point what is the problem they want to solve.

Response 1: In the introduction, a summary of SAW devices and the structures of IDT and UDT is given, although the low-loss UDT structure of SAW sensors is more easily recognized and used by scholars than the IDT structure, for the UDT structure, the structural parameters can be optimized to further reduce the losses generated, which is the main problem studied and solved in this paper. Therefore, the authors take the FEUDT structure, which is a special structure used in the UDT structure, as the research object, and use COMSOL finite element simulation software to optimize the piezoelectric substrate, forked-finger electrode shape and electrode material of this structure to derive the optimal design parameters. Assitionally based on the above optimal parameters, the law of coating film thickness is derived by analyzing the load-sensitive film of the device structure.

Round 2

Reviewer 1 Report

OK, now the manuscript is improved quite well,

please only change the first sentence in the abstract :   "To improve ....."

Author Response

Response to Reviewer 1 Comments

Point 1: please only change the first sentence in the abstract :  "To improve ....."

Response 1: In the first sentence in the abstract, the words "To improve the degree of frequency response of the SAW gas sensor for gas detection, the structure of the forked-finger transducer is analyzed and the optimal design parameters are simulated." have been changed to "In order to further improve the degree of frequency response of the surface acoustic wave (SAW) sensor for gas detection, the structure of the forked-finger transducer was analyzed, and its optimal structural parameters were simulated and designed." and marked in the text.

Reviewer 3 Report

The author said: For the UDT structure, the structural parameters can be optimized to further reduce the losses generated, which is the main problem studied and solved in this paper.   

can they provide some evidence for the advantage compared with the IDT structure in the work. 
